# Hemifusomes and interacting proteolipid nanodroplets mediate multi-vesicular body formation

Amirrasoul Tavakoli [1,3], Shiqiong Hu [1,3], Seham Ebrahim [2] ✉ & Bechara Kachar [1] ✉

The pleiomorphic structure and dynamic behavior of cellular endomembrane systems have been extensively studied using classical electron microscopy. However, fixation and staining constraints limit the in situ visualization of transient interactions, such as membrane fusion, scission, and intraluminal vesicle formation, potentially overlooking intermediate structures like membrane hemifusion. Using in situ cryo-electron tomography in four mammalian cell lines, we identify heterotypic hemifused vesicles featuring an extended hemifusion diaphragm consistently associated with a 42-nanometer proteolipid nanodroplet (PND). We designate these vesicular organelle complexes as "hemifusomes." Hemifusomes constitute up to 10% of vesicular organelles at the cell periphery but do not engage in canonical endocytic pathways. These structures exhibit diverse conformations and frequently contain intraluminal vesicles. Building on the continuum of related morphologies observed, we propose that hemifusomes serve as platforms for vesicular biogenesis, mediated by the PND. These findings provide direct in situ evidence of long-lived hemifused vesicle complexes and introduce an ESCRT-independent model for multivesicular body (MVB) formation.

The ER, Golgi, endolysosomal and secretory systems- comprising networks of structurally and functionally diverse membrane-bound organelles undergoing constant remodeling, trafficking, and recycling or degradation- are integral to cellular function and homeostasis. Central to this broad range of cellular activities are membrane fusion events required for membrane and content mixing, and membrane budding, and scission, sometimes involved in the genesis of intraluminal vesicles[1–4]. Despite significant research elucidating the molecular machineries involved in these processes[5–7], the direct, in-situ visualization and characterization of intermediate structures of membrane fusion and scission remain formidable challenges[8]. Similarly, while extensive literature exists on protein machineries required for the formation of intraluminal vesicles, the structural intermediates of membrane budding and scission events that occur during this

process remain poorly documented and inadequately understood[5,7,9]. The precise mechanisms by which membrane remodeling is coordinated to ensure the proper trafficking and turnover of membrane proteins and lipids and to maintain cellular health and function remain topics of broad interest and vigorous research. Dysregulation of membrane traffic and secretory pathways has been implicated in numerous diseases[10–12], underscoring the importance of understanding these processes for the development of targeted therapies.

Recent advances in cryo-electron tomography (cryo-ET) have enabled unprecedented visualization of cellular structures in near-native states[13–17], yielding insights into cellular structure/function relationships. The current study leverages in situ cryo-ET to uncover a class of organelles in mammalian cells, termed 'hemifusomes'. These organelles, found in all four different cell types we investigated, are

[1]Laboratory of Cell Structure and Dynamics, National Institute on Deafness and Other Communication Disorders, National Institutes of Health, Bethesda, MD, USA. [2]Center for Membrane and Cell Physiology, Department of Molecular Physiology and Biological Physics, University of Virginia, Charlotteville, VA, USA. [3]These authors contributed equally: Amirrasoul Tavakoli, Shiqiong Hu. ✉e-mail: seham.ebrahim@virginia.edu; kacharb@nidcd.nih.gov

characterized by heterotypic vesicle pairs hemifused via expanded hemifusion diaphragms (HDs), a unique membrane topology previously presumed to be too unstable for any biological function beyond serving as fleeting intermediates in membrane fusion and scission[18–20].

Hemifusomes appear in two related morphological configurations: (1) a direct hemifusion of two heterotypic vesicles; and (2) a flipped conformation where an intraluminal vesicle is hemifused to the luminal side of the membrane bilayer of a larger vesicle. In both cases, a smaller vesicle and a larger vesicle share a HD. In the case of direct hemifusomes, the topology and content of the paired heterotypic vesicles are strikingly consistent. Additionally consistent is the association of a proteolipid particle or droplet with the rim of the HD. In the case of flipped hemifusomes, more complex compound fusion configurations often arise. We explore relationships between these morphologically diverse but topologically similar conformations and their potential participation in endolysosomal functions and dynamics, including formation of intraluminal vesicles and biogenesis of multivesicular bodies.

## Results

### Observation of hemifused vesicles at the periphery of cultured cells

The periphery of vitrified mammalian cells (COS-7, HeLa, RAT-1, and 3T3 cells) of approximately 150 to 400 nm ice thickness, was surveyed using a Krios (Thermo-Fisher) cryo-electron microscope operating at 300 kV (Supplementary Fig. 1a, b). These thin cellular regions enabled visualization of membrane-bound organelles that were restricted to diameters of up to 400 nm (Fig. 1a, Supplementary Figs. 1, 2, and 3h). During a lower magnification initial survey of this region, we identified vesicles that, based on size, morphology and content, are likely to be ribosome associated vesicles (RAVs), endosomes, late endosomes, lysosomes, or multivesicular bodies (MVBs) (Fig. 1 and Supplementary Fig. 1). While vesicles at the leading edge of the cell were predominantly spherical (Figs. 1 and 2), we did observe features such as tubulation and budding, characteristic of late endosomes and lysosomes, further away from the cell periphery (Supplementary Fig. 2b).

Strikingly, we also identified closely interacting pairs of vesicles in two previously undescribed conformations: the first, a smaller vesicle appearing hemifused to the cytoplasmic side of a larger vesicle, and the second, a flipped configuration, where an intraluminal vesicle appears hemifused with the luminal side of the larger vesicle (Fig. 1 and Supplementary Fig. 1). These hemifused organelles were observed in all four cell types examined (Fig. 1a and Supplementary Fig. 1b, d, e). Surveying randomly selected ~10 μm² regions from the periphery of COS-7 cells (as in Supplementary Fig. 1b), we estimated the frequency of these hemifused vesicles to be up to 10% of the total number of membrane-bound organelles. The average number of hemifused vesicles = $0.6 \pm 0.7$ per $10~\mu m^2$; average number of other vesicles = $6.6 \pm 4.0$ per $10~\mu m^2$ ($n = 81$), with broad local variation within and between cells (Fig. 1 and Supplementary Fig. 1).

### Cryo-electron tomography of hemifused vesicles

For a more detailed analysis of morphology of the hemifused vesicles and their spatial context within the cytoplasm, we performed low-dose tilt series image acquisitions for cryo-ET, following the methodology described by Hagen et al. [21]. The workflow for this procedure is detailed in the Methods section, and representative images are shown in Supplementary Fig. 1a–c. Tomogram slices of the periphery of the cell revealed cytoskeletal components and more detailed views of vesicular elements, including clathrin-coated vesicles, endosome-like vesicles, late endosomes, RAVs, MVBs, and the previously undescribed hemifused vesicles (Fig. 1b, c and Fig. 2). Tomographic mid-cross-sections through direct and flipped hemifusomes revealed the well-defined bilayer outline of the vesicle membranes (Fig. 1b, c, and

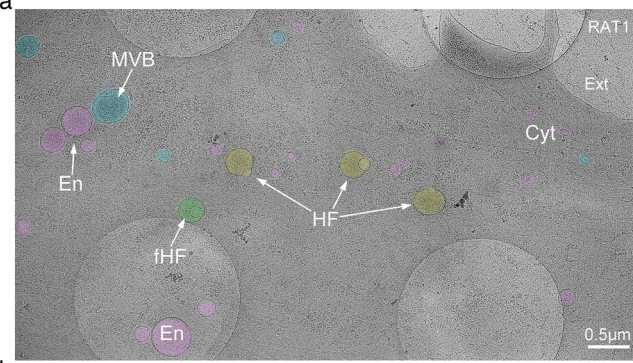
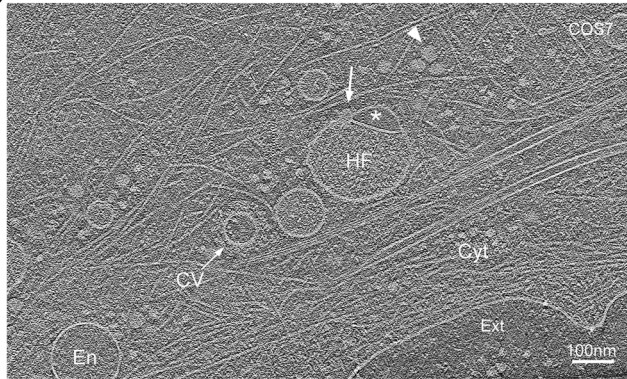
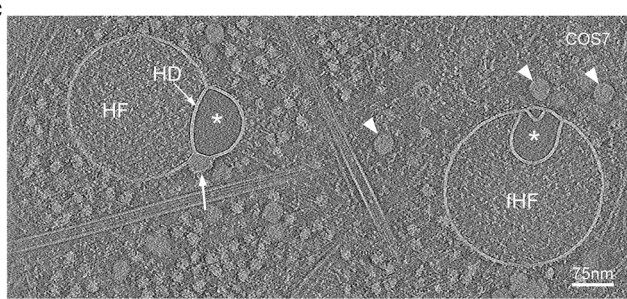

**Fig. 1 | Cryo-electron tomography observation of hemifused vesicles at the leading edge of cultured cells. a** Representative cryo-electron microscopy image of the leading edge of a RAT-1 cell cultured on cryo-EM grids. Lamellipodia and filopodia in the upper right corner delineate the cell border, separating the cytoplasm (Cyt) from the extracellular space (Ext). Vesicular organelles are highlighted in color: early endosome-like vesicles (En, pink), multivesicular bodies (MVB, blue), hemifusomes (HF, yellow), and flipped hemifusomes (fHF, green) (representative from $n = 308$ overview images before tilt series data collection). Scale bar: 0.5 μm. **b** Representative cryo-electron tomogram slice of the border of a COS-7 cell highlighting cytoskeletal components, endosome-like vesicles (En), a clathrin-coated vesicle (CV), and a hemifusome (HF). A single bilayer or hemifusion diaphragm separates the hemifused vesicles. The larger vesicle has a fine granular content, and the smaller hemifused vesicle has a smooth translucent lumen (*). The arrowhead points to the proteolipid particle and arrow points to the nanodroplet at the rim of the hemifusion diaphragm (representative from at least 10 tomograms). Scale bar: 100 nm. **c** Tomographic mid-cross-section through a direct (HF) (from $n = 88$ tomograms) and a flipped hemifusome (fHF) (from $n = 48$ tomograms) showing the well-defined bilayer outline of the vesicle membranes (from $n = 308$ tomograms). A single bilayer or hemifusion diaphragm (HD) separates the hemifused vesicles. Arrowheads point to the proteolipid particle, as well as similar proteolipid particles seen free in the cytoplasm. Scale bar: 75 nm.

Supplementary Fig. 1c). The tomograms also confirmed the presence of a shared bilayer at the interface of the vesicle pair (Fig. 1b, c, Supplementary Fig. 1c, and Supplementary Movies 1 and 2), characteristic of an extended HD, a fusion intermediate theorized[18–20,22] and observed in synthetic[23] or cell-free[8,14,24] systems, but rarely observed in intact cells[25,26].

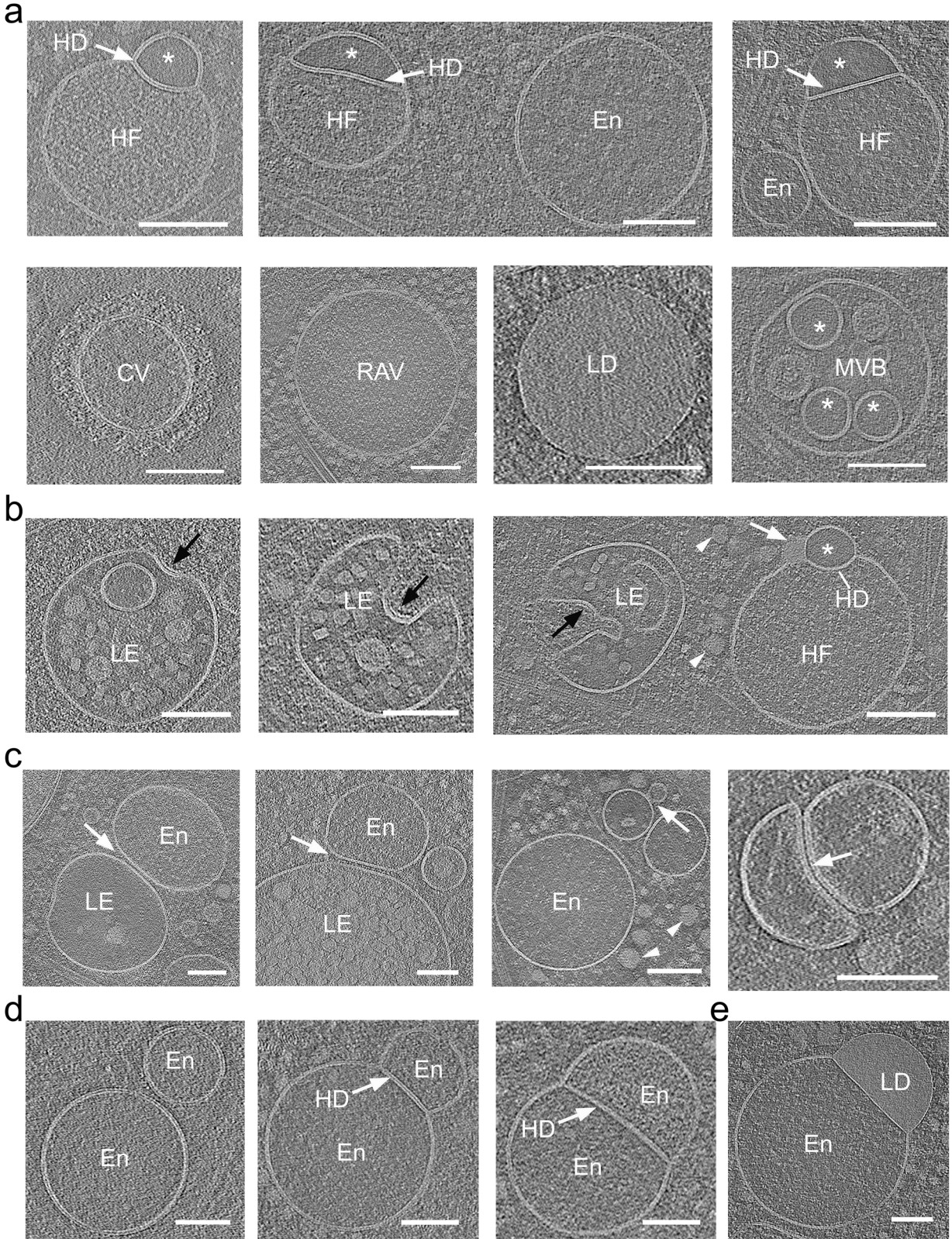

While the lumen of the larger vesicles displayed a fine granular texture, comparable to the lumen of endosome-like vesicles and RAVs, the smaller vesicles in these pairs consistently exhibited "translucent" content devoid of visible differential electron scattering or phase contrast typical for granular or particulate material (Figs. 1b, c, 2, 3), we posit that the translucent luminal content likely reflects a protein-free or very dilute aqueous solution.

We designate the term "direct hemifusomes" or simply "hemifusomes" to the hemifused vesicles where the smaller translucent vesicle is on the cytoplasmic side of the membrane of the larger vesicle, and "flipped hemifusomes" to the conformation where the translucent vesicle is hemifused to the luminal or exoplasmic side of the larger vesicle membrane (Fig. 1c and Supplementary Movies 1–3). Additionally, 3D tomogram reconstructions frequently revealed a dense or

**Fig. 2 | Hemifusome luminal content is distinct from other membrane-bound organelles. a** Representative cryo-electron microscopy images of various membrane-bound organelles within the endo-lysosomal system. Endosome-like vesicles (En), late endosomes or endolysosomes (LE), a clathrin-coated vesicle (CV), ribosome-associated vesicles (RAV), multivesicular body (MVB), lipid droplet (LD), and hemifusomes (HF) are identified. The distinct luminal content of each organelle is visible, with the smaller hemifusome vesicles (*) consistently showing a unique light, smooth, and particle-free luminal content compared to other organelles. Similar smooth lumen vesicles were found only inside some MVBs (*). Hemifusion diaphragm (HD) is highlighted with white arrows (representative image from at least 10 tomograms). **b** Series of endolysosomes or late endosomes (LE) at various initial stages of inward budding of a vesicle obtained from tomographic slices. The

last panel shows a side-by-side comparison of an endolysosome and a hemifusome (HF). Black arrows point to a distinct surface protein complex, likely ESCRT, at the inwardly curved portion of the endolysosomal membrane. White arrowhead points to proteolipid nanodroplets in the cytoplasm and white arrow points to the nanodroplet at the rim of the hemifusion diaphragm (representative image from at least 10 tomograms). **c** Tomogram slices showing endosomal-like vesicles (En) and late endosomes or endolysosomes (LE) vesicles adhered or docked to each other (white arrows). Representative images from at least 10 tomograms. **d** Tomogram slices showing homotypic and heterotypic hemifused endosome-like vesicles (En) sharing an extended hemifusion diaphragm (HD) (white arrow). Right panel: a hemifused vesicle and lipid droplet (LD). Note the granular texture of the lumen of all vesicles (representative images from at least 10 tomograms). Scale bars: 100 nm.

phase-dark particle, or nanodroplet, integrated into the bilayer at the margin of the HD (arrows in Fig. 1b, c). Seemingly membrane-less particles of similar size and overall appearance can be observed in the cytoplasm surrounding the hemifusomes in the tomogram slices (indicated by arrowheads in Fig. 1b, c).

Our observation of hemifusomes in four different cell lines originating from various species and tissues and frozen as close as possible to their native state suggests that they may be common components of the cell periphery in a wide range of cells and tissues. Additionally, a review of archival transmission electron microscopy images of plastic-embedded thin sections of conventionally prepared inner ear tissue from our lab revealed structures resembling hemifused vesicles within the endosomal compartment of epithelial cells[27] (Supplementary Fig. 2b).

### Comparison of the hemifusome luminal content to the content of other membrane-bound organelles

The area being imaged in this study is at the leading edge of the cell, which is in the range of 150–400 nm thick. The membrane organelles in this region are thus all smaller than 400 nm in diameter (most between 100 and 300 nm). Additionally, these organelles are in constant and rapid motion, as evident in a high-performance phase-contrast live imaging movie (time-lapse of short exposures) of the COS-7 cell periphery (Supplementary Movie 4). The sub-diffraction vesicles in our tomograms correspond to the smallest and faintest structures in the movie, near the cell edge, undergoing fast movements (not the more birefringent structures). It is thus important to note that together, the size and speed of these organelles preclude the ability of their characterization by light-microscopy.

To explore the nature of the direct hemifusome, we compared its morphology and the appearance of its luminal content with other known vesicular organelles in the surrounding cytoplasm (Fig. 2a, b). As stated above, the lumen of the larger vesicles in the heterotypic hemifused pair displayed a fine granular texture, comparable to the lumen of RAVs and endosome-like vesicles (Fig. 2a, b). However, the unique smooth and translucent appearance of the luminal content of the smaller vesicle (asterisks in Fig. 2a, b) did not match the texture or electron density of the lumen of any of the other vesicular organelles, including endosome-like vesicles, clathrin-coated vesicles, RAVs, lipid droplets (Fig. 2a), or various conformations of endolysosomes (Fig. 2b). The only vesicles in the 308 tomograms we acquired with comparable content were some of the intraluminal vesicles in MVBs (asterisks in Fig. 2a, lower panel).

We also observed (at least 10 examples in 308 tomograms) other vesicles (Fig. 2c, d) and lipid droplets (Fig. 2e), either docked (Fig. 2c) or hemifused (Fig. 2d, e), in line with established models of docking[23,28–30] and fusion that lead to the formation of late endosomes or the delivery of endosomal cargo to lysosomes[4,31]. Notably, we did not observe any translucent vesicles that were either free or docked to vesicles other than hemifusomes. Further, we also identified hemifusomes in actin-rich cellular regions largely devoid of membrane-bound organelles (Supplementary Fig. 1f) where the chances of vesicle

encounter, docking, and subsequent fusion to form a hemifused pair would be low. Together, these findings led us to question whether hemifusomes might be formed by alternative mechanisms to canonical vesicle fusion.

### Morphological variation of direct hemifusomes

Representative close-up views of the rim of the hemifusome HD, where the bilayers of the two vesicles and the HD meet, enabled us to confirm that the cytoplasmic leaflets of the bilayers of the two interacting vesicles were contiguous (Figs. 1c, 2, and 3). The leaflets of the HD bilayer are comprised of the exoplasmic leaflets of the two interacting vesicles, as expected for hemifused vesicles. These bilayer arrangements are depicted in Fig. 3c and e. Measurements of the thickness of the bilayer of each hemifusome vesicle, and the shared HD, were comparable to typical membranes at ~4 nm (Supplementary Fig. 4g and h). The paired vesicles forming the hemifusome can vary widely in both individual and relative sizes. Additionally, even when the paired vesicles are of comparable-size, their HDs exhibit variability in radius and curvature indicating diverse degrees of radial expansion of the area of hemifusion between vesicles (Fig. 3 and Supplementary Fig. 4).

Independent of how the hemifusomes are formed, radial expansion of HDs must require adhesive forces capable of deforming vesicles and overcoming the ensuing increase of internal pressure and membrane tension as the surface-to-volume ratio changes for both vesicles. The observation of the HD bulging into the larger vesicle is consistent with intrinsic pressure differential that occurs when the HD expands; namely, the smaller vesicle experiences higher pressure due to the faster rate of change in surface-to-volume ratio. However, intriguingly, HDs could also be found, albeit less frequently, with no curvature or even bulging towards the smaller vesicle (Fig. 3a). Local changes in the composition and biophysical properties of the bilayers are expected as the angle/curvature of the monolayers change at the rim of the HD[22,32–34].

In addition to the biophysical properties of the membrane[20] and other factors intrinsic to the hemifusome complex, the shape of the complex could also be influenced by local physical constraints. We observed hemifusomes deformed by surrounding actin filaments and microtubules (Supplementary Fig. 3a, b) or by constraints imposed by proximity to the plasma membrane (Supplementary Fig. 3b, c) as the hemifusome squeezed through the thinner regions at the cell's edge. These varying conformations illustrate the compressibility and deformability of the hemifusomes and how internal and external forces or constraints (Fig. 3f and Supplementary Fig. 3) potentially contribute to their overall shape, expansion of the HDs, and the angles and curvature of the interacting membranes at the rim of the HDs.

### Emergence of intramembrane lens-shaped structures in hemifusomes

The average radius for the larger vesicle in the hemifusome in the areas we surveyed was measured to be ~299.3 ± 96.2 nm ($n = 50$), and for the HDs of the same hemifusomes was measured to be 158.4 ± 60.9 nm ($n = 50$) (Fig. 3d). This is an order of magnitude larger than the

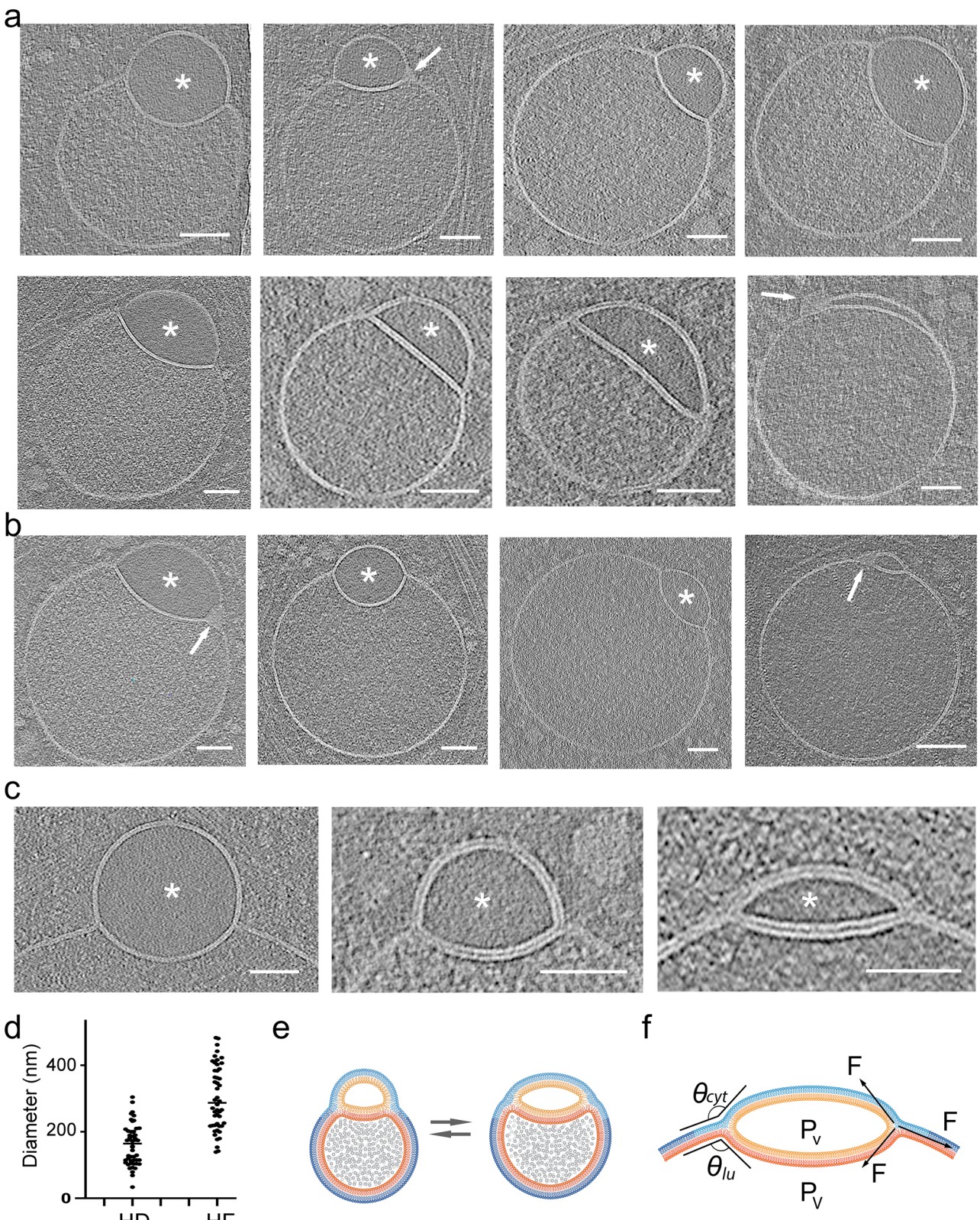

transient ~10 nm HD estimated to exist in canonical membrane fusion events[14,18,35,36]. This finding implies that the expansion of the hemifusome HD is energetically favorable, and large HDs (ratio of HD to hemifusome diameters is shown in Supplementary Fig. 4) are likely stable with an extended lifespan.

Indeed, we often observe hemifusome HDs comparable in size to the remaining membrane of the smaller vesicle (Fig. 3b). In this

configuration, the entire inner leaflet and contents of the smaller vesicle are fully encapsulated within the bilayer of the membrane encompassing the larger vesicle. This symmetric intra-bilayer structure (Fig. 3e) has been identified in silico simulations as a long-lived, lens-shaped product of hemifusion between a lipid vesicle and a planar lipid bilayer, referred to as 'dead-end' hemifusion[18]. It is particularly exciting that we observe such long-lived, lens-shaped structures within a

**Fig. 3 | Range of morphological appearances of direct hemifusomes. a** Cryo-electron tomography mid-cross-section slices of various direct hemifusomes highlighting the variability in sizes and shapes across and within each hemifusome pair. Note the smooth appearance of the smaller vesicle of the hemifusome (*). Hemifusomes also show variability in their hemifusion diaphragm diameter and curvature. Arrows point to proteolipid particles lodged in the hemifusome at the rim of the hemifusion diaphragm where it meets with membranes of the two vesicles (*n* = 88 from 308 tomograms). **b** Hemifusomes showing deformation of the smaller vesicle with the expansion and curvature of the hemifusion diaphragm, resulting in a cross-sectional view that resembles a lens-shaped vesicle. In this specific configuration, the entire inner leaflet as well as the content of the smaller vesicle becomes fully embedded within the bilayer of the membrane containing the larger vesicle (representative images from at least 10 tomograms). **c** Close-up views of the hemifusion diaphragm and degrees of flattening of the smaller vesicle (*) to form the lens-shaped structure embedded in the bilayer (representative images from at least 10 tomograms). Scale bars: 50 nm. **d** Diameters of the hemifusion diaphragm (HD), mean= 158.4 ± 60.9 nm, and the larger vesicle of the hemifusome (HF), mean = 299.3 ± 96.2 nm. *n* = 50 hemifusomes. **e** Diagram illustrating the hemifused configuration and possible interconversion between hemifusome conformations. **f** Diagram illustrating some of the forces at play to modulate hemifusome angles between the membranes and overall shape. θ = angle of the cytoplasmic (*cyt*) and luminal (*lu*) leaflets of the bilayer at the point of junction with the hemifusion diaphragm.; P, internal pressure of smaller (v) and larger vesicle (V), F, membrane tension vectors.

complex biological membranous organelle. Consistent with the in-silico simulations, this lens-shaped conformation is observed in hemifusomes containing very small, translucent vesicles (Fig. 3b, c), where the larger vesicle closely approximates the flatness of a planar lipid bilayer.

## Parallels between direct and flipped hemifusomes

In closely examined 308 tomograms, we were able to identify clearly 88 direct hemifusomes (representative examples in Fig. 3) and 48 hemifusomes in the flipped configuration (representative examples in Fig. 4). Based on the conserved topological features of direct and flipped hemifusomes, we hypothesize that they may represent different conformations of the same organelle, potentially transitioning from one form to the other.

Among the rounded hemifusomes, we observed a continuum of conformations (Fig. 4 and Supplementary Fig. 4) that appear to correspond to various stages of HD expansion and intraluminal budding of the lens-shaped structure. Similarly, based on the range of morphologies of the flipped hemifusomes, we identified two modes of intraluminal budding. In the first (Fig. 4a), the smaller vesicle of the hemifusome pair is well-rounded, optimizing the volume-to-surface area ratio. In the second (Fig. 4b, c), the budding vesicle exhibits an elongated shape. We postulate that in both scenarios, the original hemifusome HD, with two exoplasmic leaflets, curves and expands to form the luminal hemifused vesicle, while the cytoplasmic portion of the bilayer shared by both vesicles decreases in radius (Fig. 4a–c) transforming into an external hemifusion diaphragm (EHD), which is ultimately comprised of both an exoplasmic and a cytoplasmic leaflet (Fig. 4f). The consistent presence of the EHD is a critical feature that distinguishes the luminal budding vesicles in the flipped hemifusome from the canonical ESCRT-based budding and formation of intraluminal vesicles[5]. In the ESCRT model, the budding portion of the membrane forms an omega shape, with the cavity open to the cytoplasm (Fig. 2b) until the neck of the forming vesicle undergoes scission. Conversely, in the hemifusome, there is invariably an HD separating the cytoplasm from the lumen of the inwards-flipping vesicle (Fig. 4). Furthermore, in the ESCRT-based model, the content of the cavity corresponds to the portion of cytoplasm being captured (Fig. 2b), while in the reverse fusion process, the texture of the luminal content is smooth and lighter in contrast (Fig. 4), consistent with the "translucent" content of the smaller or lens-shaped vesicle of the hemifusome (asterisks in Fig. 3).

Intriguingly, the width of the smaller EHDs appear to be less than double the thickness of the bilayer (Fig. 4c–e), akin to the reported dimensions of the canonical fission intermediate "stalk" structures[14,20,22,36,37]. While we did not directly visualize the merging of the luminal membranes due to the resolution limitations of our tomograms, the point-like contact of the membranes as shown in Fig. 4d, e is consistent in size to that of a stalk. This putative conversion of an HD into a stalk is akin to a reverse canonical fusion process[18,20,38] where the stalk precedes the formation of an HD. The diagram in Fig. 4f illustrates the range of conformations observed, and our proposed model showing how these conformations could fit within a progressive intraluminal budding of the translucent vesicle, with a concomitant reduction of the EHD until a stalk is formed. Like the scission of the intraluminal vesicle proposed in ESCRT-based intraluminal vesicle formation[5], it is plausible that the scission of the flipped hemifusome stalk then results in the pinching off of the vesicle and the formation of a free intraluminal vesicle.

## Exploring the relationship between hemifusomes and endosomes using gold nanoparticles

Given the similarity in size and content between the larger compartment of the hemifusome and similarly sized endosomal vesicles observed in our tomogram slices (Supplementary Fig. 2a), we sought to explore the potential relationship between hemifusomes and the endosomal pathway by employing functionalized gold nanoparticles as tracers of endocytosis. Endosomes represent population of vesicles with remarkable variation in both their source and trajectories[39]. They originate largely from the plasma membrane[40,41], but also form through the fusion of vesicles derived from other intracellular compartments, such as the trans-Golgi network[29]. During receptor-mediated endocytosis, specific ligands bind to their corresponding receptors on the plasma membrane, and these receptor-ligand complexes are subsequently internalized into subsets of endosomes[31,42]. This mechanism ensures that each endocytic pathway selectively internalizes specific cargo based on receptor-ligand interactions[42,43].

To investigate whether hemifusomes participate in the endocytic uptake pathway, we used three types of functionalized gold nanoparticles as tracers. These nanoparticles varied in size (5 nm and 15 nm) and surface functionalization, including physisorbed ferritin (Luna Nanotech), covalently bound ferritin, and a proprietary negatively charged polymer-coated nanogold (NanoPartz). After exposing the cells to the gold nanoparticles for various durations (ranging from 30 min to 3 h), we observed their localization within coated pits, coated vesicles, endosomes, and late endosomes (Fig. 5 and Supplementary Fig. 5). Strikingly, we were unable to detect gold nanoparticles within either vesicle of the hemifusomes (Fig. 5). Notably, we also observed many endosomes and late endosomes that did not contain gold (Fig. 5), consistent with the diversity of endocytic pathways[44–47]. Taken together, the results of our uptake experiment suggest that hemifusomes are not part of the transferrin endocytic pathway, but neither conclusively establish nor dismiss the possibility that hemifusomes represent a distinct subset of endosomes[39,42].

## Proteolipid nanodroplet at the rim of the hemifusion diaphragm

As previously noted, tomographic analyses of hemifusomes reveal a single dense structure embedded within the hydrophobic interior of the bilayers at the three-way junction of the HD and the two heterotypic vesicles (Figs. 1b, 1c, 3a, 3b, 6a, 6b, and Supplementary Fig. 1c). Detailed views show this dense structure,

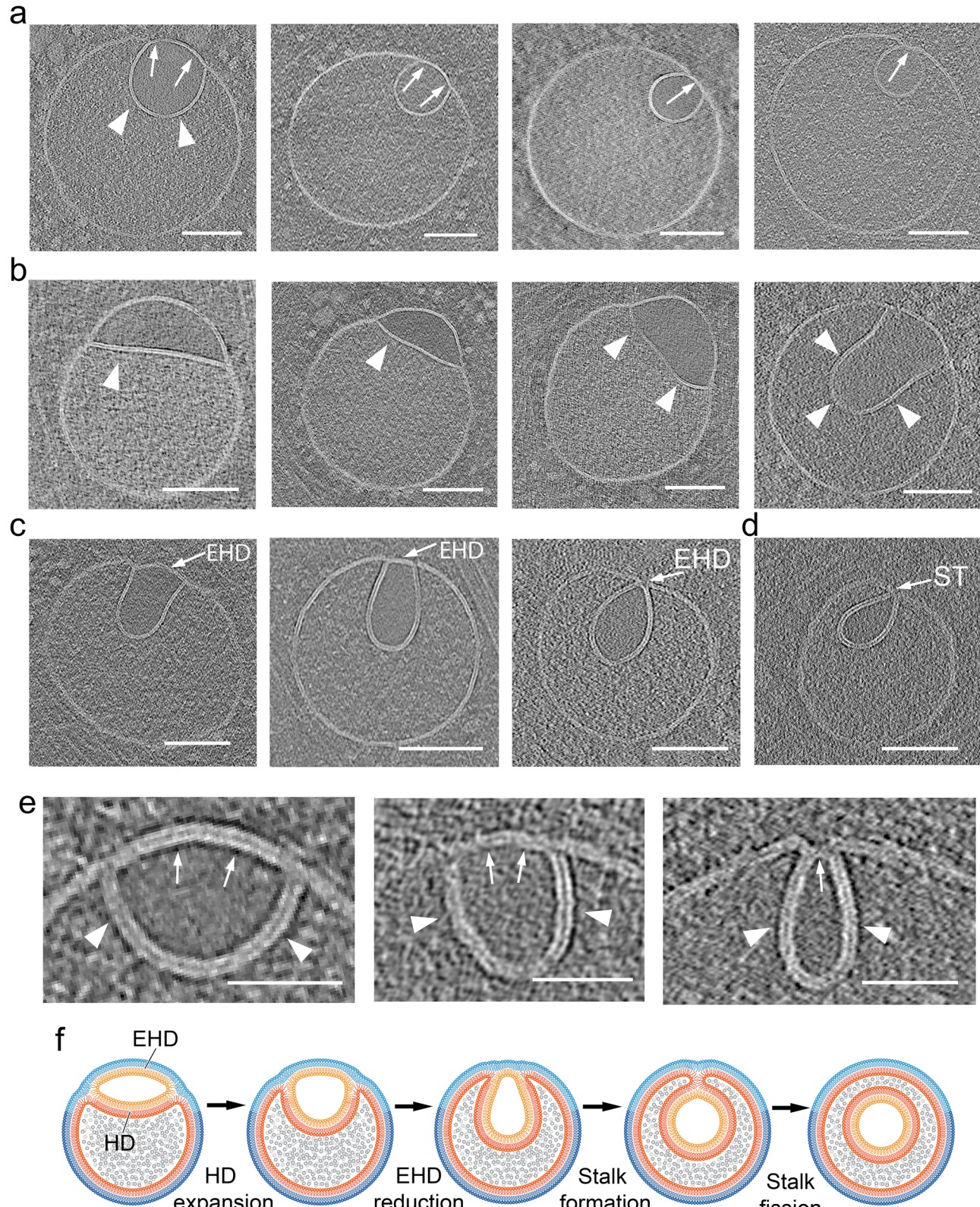

**Fig. 4 | Range of morphological appearances of flipped hemifusomes and the proposed progression to form an intraluminal vesicle. a** Tomographic slices of various flipped hemifusomes (fHF) highlighting the variability in sizes and shapes across and within each hemifusome pair (*n* = 48 from 308 tomograms). The budding vesicle exhibits a circular shape. **b–d** Various degrees of curving and budding of the hemifusion diaphragm. The budding vesicle exhibits an elongated shape. In both circular and elongated scenarios, the intraluminal portion of the hemifused diaphragm expands, while the cytoplasmic side of the vesicle decreases in radius and forms an external segment of the bilayer shared by both vesicles. During this proposed process, the cytoplasmic side of the lens-shaped structure transforms into an external hemifusion diaphragm (EHD), which reduces radially to form a stalk-like structure (ST) (representative images from at least 10 tomograms). **e** Close-up views of the curving and expansion of the hemifusion diaphragm (HD, arrowheads) and radial reduction of the EHD (arrows) (representative images from at least 10 tomograms). **f** Diagram depicting our proposed model, illustrating the progressive rounding and expansion of the hemifusion diaphragm (HD) and reduction of the EHD to form a stalk and ultimately scission to form an intraluminal vesicle. Scale bars: 50 nm.

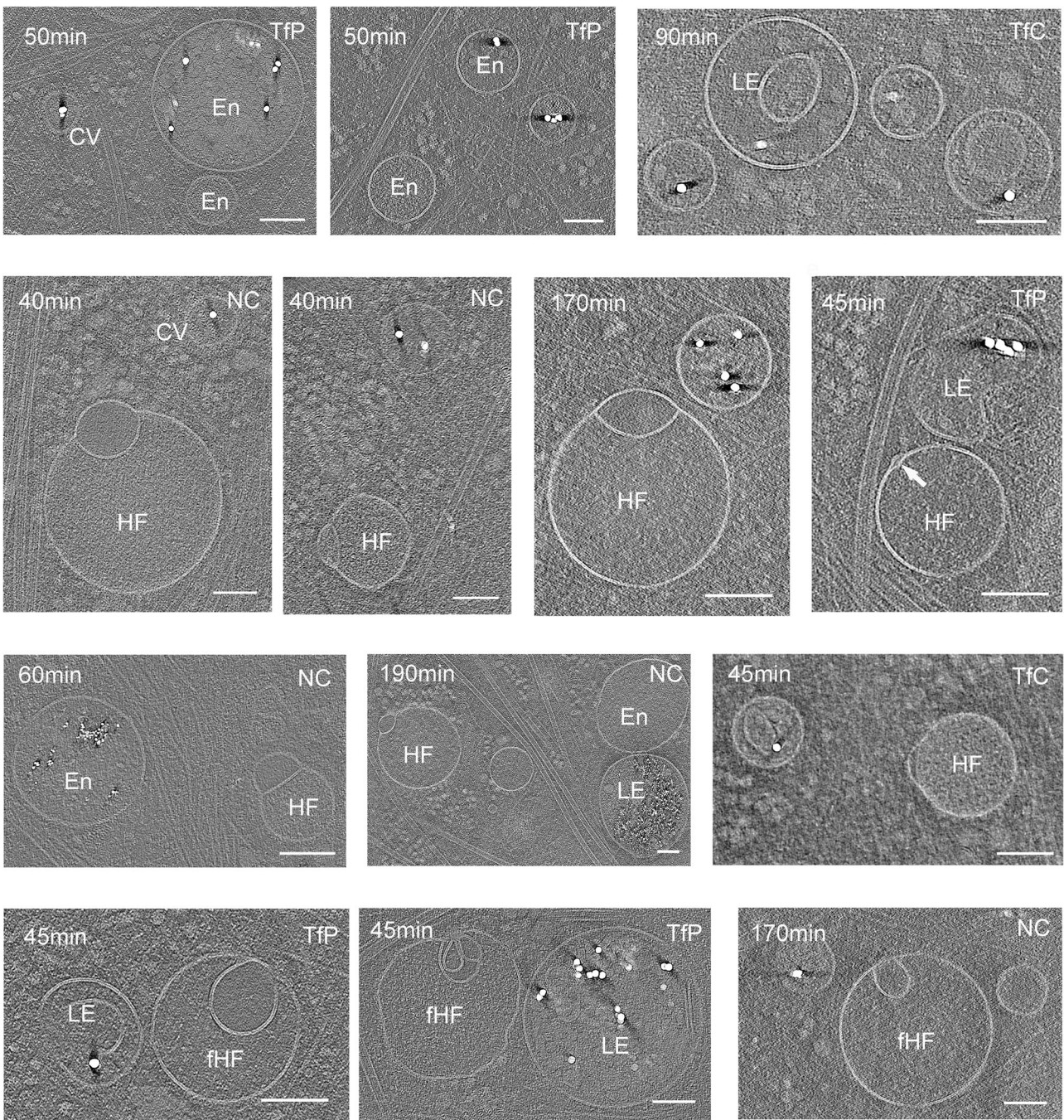

**Fig. 5 | Hemifusomes are not part of the uptake and cargo transfer of endocytosed nanogold particles.** Tomographic slices of pulse-chase experiments showing the distribution nanogold particles of various surface-functionalization and sizes (small particles = 5 nm and large particles = 15 nm) in clathrin-coated pits or vesicles (CV), endosome-like vesicles (En), and late endosomes or endolysosomes (LE), but absent from either vesicle compartment of the hemifusomes (HF) or flipped hemifusomes (fHF). TfP gold nanoparticles with transferrin physiosorbed, TfC gold nanoparticles with transferrin covalently attached, and NC gold nanoparticles with slightly negatively charged non-reactive polymer. Time of incubation with the gold nanoparticles is shown in minutes. Scale bars: 100 nm. (representative images from at least 5 tomograms from at least two repeats of each experiment).

or nanodroplet, interfacing with the hydrophobic side of the exoplasmic leaflets of the hemifused vesicles and the HD (insets and close-up views in Fig. 6a, b), suggesting it contains hydrophobic components, likely lipids. The cytoplasmic surface and interior of the nanodroplet contain particulate structures (insets and close-ups in Fig. 6a, b), likely proteins, leading us to conclude that these nanodroplets are of mixed proteolipid composition.

Determining the frequency and distribution of nanodroplets around the entire HD is challenging due to the inherent missing wedge limitation[48] (Supplementary Fig. 3i, j) of cryo-ET. Typically, the best

tomograms cover about one-third of the hemifusome circumference, with the top and bottom thirds of the hemifusome missing from the tomogram (Supplementary Fig. 3i, j). Consequently, even optimally oriented tomogram slices might miss these structures. Consistent with the estimation of one nanodroplet per hemifusome, a distinct nanodroplet was observed in about half of the ~300 hemifusome tomograms examined, with no instances of two or more nanodroplets associated with a single HD.

Strikingly, similar nanodroplets are observed embedded within the hydrophobic interior of the bilayer of single vesicles (Fig. 6c–e).

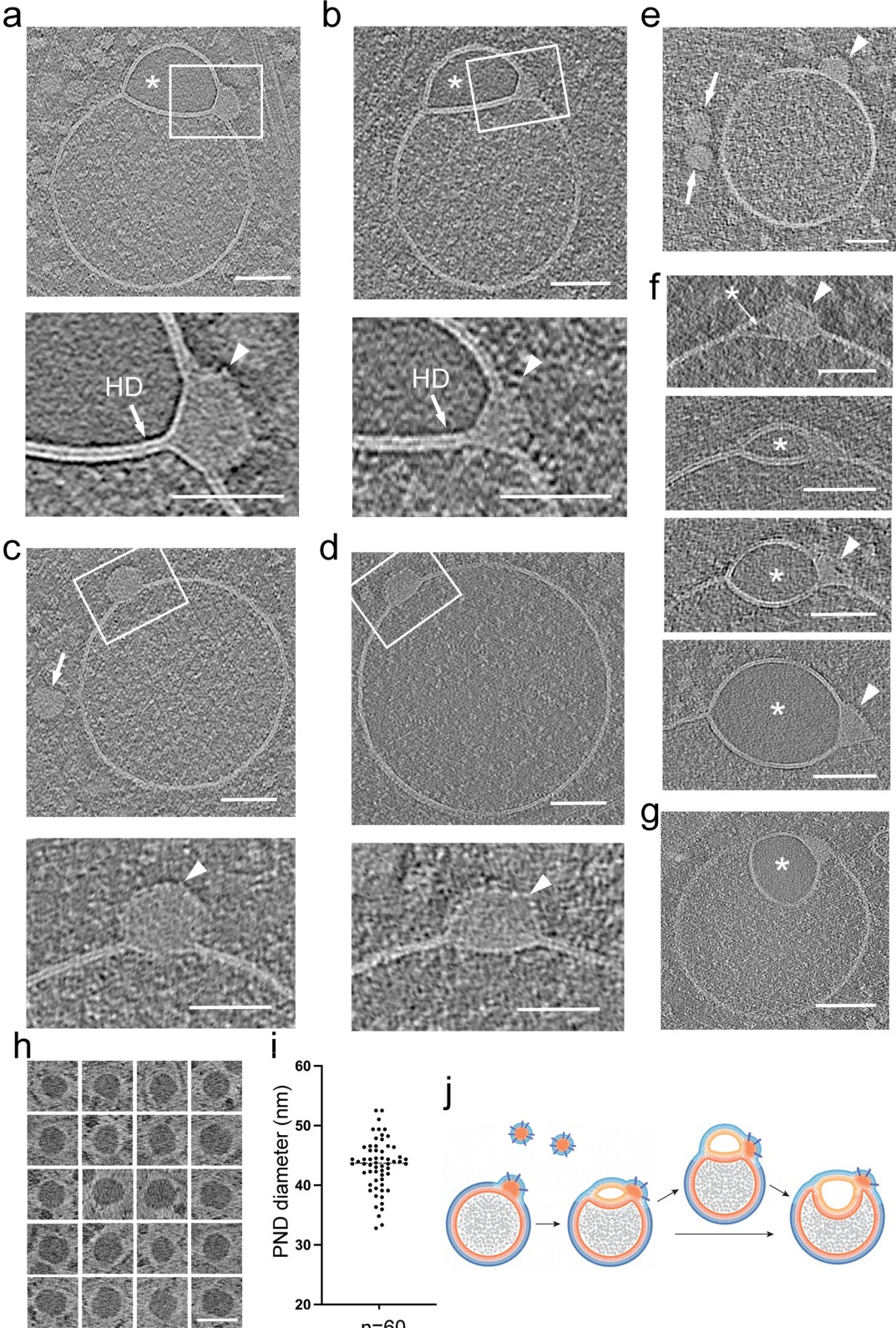

These lensed nanodroplets interface with the hydrophobic side of the exoplasmic leaflet of the vesicle bilayer, while the cytoplasmic boundaries are contiguous with the cytoplasmic leaflet of the vesicle bilayer (insets and close-ups in Fig. 6c, d).

We conducted a detailed examination of the diverse hemifusome and flipped hemifusome morphologies documented in our tomograms (Fig. 6f, g). Notably, we observed hemifusomes containing translucent vesicles, varying from minimal-sized pockets or cysts encapsulated within the bilayer adjacent to the nanodroplet, to progressively larger translucent hemifused vesicles in both direct hemifusome (indicated by asterisks in Fig. 6f) and flipped hemifusome configurations (asterisks in Fig. 6g). This observation raises the possibility that the insertion of the nanodroplet into endosomal or endosome-like vesicles might trigger the initiation, formation, or

**Fig. 6 | Proteo-lipid nanodroplet associated with hemifusomes are located at the rim of the hemifusion diaphragm. a, b** Tomographic slices and corresponding close-up views of hemifusomes (HF) with a prominent proteolipid nanodroplet (PND) at the rim of the hemifusion diaphragm (HD). Asterisk marks the smooth lumen of the smaller vesicle. Lower panels are close-up views of the area marked with a white rectangle showing the PND embedded in the hydrophobic core of the vesicle membrane and a series of particulate structures on its outer limit. These structural features reinforce the view that PNDs contain lipids and proteins (representative images from at least 10 tomograms). **c, d** Tomographic slices and corresponding close-up views of PNDs fused to individual vesicles. Arrows point to a PND free in the cytoplasm. Lower panels are high magnifications of the boxed regions, showing the PND encapsulated within the vesicle lipid bilayer. The outer leaflet enveloping the PND is decorated with protein particles (arrowheads). (representative images from at least 10 tomograms). **e** More examples of PNDs of uniform size and texture, within the cytoplasm (arrows) and fused to a vesicle (arrowhead). (representative images from at least 10 tomograms). **f** Mid-cross-section of tomograms of a series of different hemifusomes showing a range of increasing size vesicles with a smooth lumen (asterisks) formed next to the PND site (arrowheads), suggesting that the smaller vesicle of the hemifusome may be forming by a de novo vesiculogenesis process. (representative images from at least 10 tomograms). **g** Tomogram slice of a flipped hemifusome showing the PND embedded at the three-way juncture of the hemifusion diaphragm and the membrane of the two cytoplasmic vesicles (representative image from at least 10 tomograms). **h** Panel of PNDs found free in the cytoplasm in the vicinity of hemifusomes. Image contrast in this image is reversed to its original gray scale to show that the PNDs are electron dense or have a phase dark appearance. Scale bar **a–h**: 50 nm. **i** Plot of the diameter distribution of PNDs. The average diameter was calculated to be 42.4 ± 4.3 nm ($n = 60$ from 5 tomograms). **j** Diagram illustrating the association of PNDs to cytoplasmic vesicles to form a hemifusome and the progression to a flipped hemifusome.

stabilization of a vesicle hemifused to the parent vesicle to form the hemifusomes. The images, which resemble a process of membrane blistering or vesiculogenesis, suggest a mechanism involving the import of water into the lumen, potentially explaining the translucent nature of this extended compartment within the hemifusome.

The diameter of these nanodroplets, measured in tomogram slices through their mid-portion, was calculated to be 42.4 ± 4.3 nm ($n = 60$). Their size, electron density, and texture are comparable to similar particles observed in the surrounding cytoplasm (Figs. 6c, e, as well as 1b, c). Figure 6h shows a panel of representative cytoplasmic nanodroplets. The contrast in this figure is inverted to highlight that the content of nanodroplets is electron-dense or phase-dark compared to the surrounding cytoplasm and resembles that of lipid droplets (Supplementary Fig. 5b), except that they are not limited by a lipid leaflet but rather a proteinaceous coat. Based on these features, which support the idea that the nanodroplets are proteolipidic in nature, and their nanoscale size, distinct from lipid droplets (Fig. 6i), we designate these particles as proteolipid nanodroplets (PNDs).

Our findings that the PND is associated with free vesicles and with hemifusomes of various configurations lead us to postulate a model for PND-based vesiculogenesis as a mechanism of formation of hemifusomes. In this model we hypothesize that PNDs, initially free in the cytoplasm, attach to yet to be identified cytoplasmic vesicles. At the site of interaction, the PND contributes lipid and protein building blocks for the initiation of the vesicle lensed within the bilayer, which grows to form the translucent vesicle characteristic of hemifusomes (Fig. 6j).

### Compound hemifusomes as hubs for the formation of complex multivesicular bodies

One-quarter of the total number of hemifusomes, both direct (Fig. 7a) and flipped (Fig. 7b), identified showed one or more additional vesicles in a hemifused conformation with either or both vesicles of the initial hemifusome pair (88 direct, 48 flipped hemifusomes, and 42 compound out of 178 total hemifusomes; Fig. 7, Supplementary Figs. 6 and 7). The occurrence of these compound hemifusomes further demonstrates the unexpected longevity of the HDs within the hemifusomes. We also observed several instances of hemifusomes with additional PNDs embedded in their membranes. Based on our hypothesized model of PNDs as hubs for vesicle biogenesis (Fig. 6), we propose that these additional PNDs are likely sites for the initiation of compound hemifusomes.

Multiple hemifusion events coalesce to form very complex compound structures, as shown in Fig. 7c. Some of these structures harbor a range of conformations of the direct and flipped hemifusomes, suggesting that the mechanism of addition of hemifused vesicles is stochastic in time and location within the hemifusome complex (Fig. 7c). We speculate that compound hemifusomes, followed by the flipping of the vesicle into the luminal side of the larger vesicle and subsequent scission, may provide an alternative path to the formation of intraluminal vesicles (Fig. 7c, d). One relevant observation is that the outer vesicle in the hemifusome often shows a subtle crenation (Supplementary Fig. 7c, d), suggestive of reduced turgor. This reduction in turgor would facilitate the inward budding of the lens-shaped vesicle postulated in our proposed model.

## Discussion

Through the application of cryo-ET to cultured mammalian cells, we have identified a previously unrecognized vesicular organelle complex with a unique membrane topology. This complex, which we have termed 'hemifusome', consists of hemifused heterotypic vesicles sharing a large (on average ~160 nm) hemifusion diaphragm (HD). The presence of such large, long-lived HDs is particularly surprising, given the widely accepted view that HDs are small (less than 10 nm), unstable, and typically occur as transient intermediates during rapid vesicle fusion processes leading to vesicle content mixing[20,22,36]. A second intriguing feature of hemifusomes is the consistent presence of the distinct ~42 nm proteolipid nanodroplet (PND) embedded in the membrane of the hemifused vesicles at the rim of the HDs. This localization suggests a role for the PND in the formation, stabilization, or expansion of the hemifusion interface or in hemifusome biogenesis and dynamics.

The paired vesicles in each hemifusome are heterotypic in both size and luminal content. Hemifusomes present in two distinct conformations: a direct hemifusome, where the smaller vesicle is hemifused to the cytoplasmic side, and a flipped hemifusome, where the hemifused vesicle is internal and fused to the luminal side of the parent vesicle. Given their distinctive topology and variety of conformations, we predict that hemifusomes may play roles in protein and lipid sorting, recycling, and the formation of intraluminal vesicles.

One possibility for hemifusome formation is through the hemifusion of two pre-existing vesicles. This hypothesis is supported by the occasional observation of docked and hemifused endosomal-like vesicles with expanded HDs (Fig. 2c, d). However, the smaller vesicles in the hemifusome consistently contain a translucent luminal content, distinct from the appearance of the luminal content in all other vesicular organelles (including endosomes, lysosomes, and RAVs). In fact, in all 308 tomograms, we did not observe similar individual translucent vesicles in the cytoplasm. The only vesicles we observed with similar translucent content were some of the intraluminal vesicles in MVBs. The absence of free translucent vesicles in the cytoplasm or docked to other vesicles challenges the hypothesis that hemifusome formation occurs through canonical SNARE-mediated vesicle fusion.

The range of hemifusome morphologies observed in our in situ cryo-ET data—spanning from slightly to highly deformed hemifused vesicles sharing an expanded HD—has not been previously described in situ. These morphologies align with stages of HD remodeling and

expansion predicted by mathematical models, synthetic lipid systems, and in vitro observations of reconstituted systems[18], where it has been suggested that a growing HD experiences high tension and may rupture if subjected to osmotic pressure or high-curvature membrane stresses[22,33,49]. Detailed in situ analyses of lipid systems indicate that the HD can stabilize into a "lens-shaped" complex, known as "dead-end hemifusion"[18,22], which encapsulates a vesicle within a bilayer. Our observation of such structures in cells confirms that this phenomenon occurs in biological membranes in intact cells, and potentially serves a biological function beyond acting as an intermediate of fusion.

Based on our observations, we propose that in hemifusomes, the lens-shaped structures are long-lived and can be remodeled beyond the theoretical dead-end formulation, evolving into intraluminal vesicles. This process may involve the progressive intraluminal budding of the smaller vesicle through a mechanism distinct from the canonical ESCRT pathway. The presence of an extended HD, as opposed to the "omega"-shaped budding that captures a portion of the cytoplasm, as described in ESCRT-mediated intraluminal vesicle formation, highlights the unique nature of hemifusome-mediated vesicle internalization. Given the consistent presence of a PND free in the cytoplasm, attached to endosome-like vesicles, and at the rim of the hemifusome HD, we hypothesize that PND are hubs for hemifusome formation, acting as pockets of membrane building blocks that utilize endosomal vesicle bilayers as sites and templates to generate new hemifused vesicles (Fig. 6). We further argue that this PND-dependent process, which we term de novo "vesiculogenesis," represents an alternative to canonical ESCRT-based vesicle budding and intraluminal vesicle formation (Fig. 7). Nevertheless, further experimental validation is necessary to substantiate the proposed pathways.

The ESCRT-model of intraluminal vesicle formation presents with several challenges[5,9], particularly regarding the formation of MVBs containing multiple and diverse[44] intraluminal vesicles. Generating sufficient membrane area in these cases would require a large supply of lipids, which is unlikely to be sourced from a single vesicle through repeated ESCRT-based internal budding and scission without an additional lipid source. Moreover, while there is evidence supporting the role of ESCRT filaments in promoting membrane budding and scission, direct evidence explaining how ESCRTs facilitate the formation of numerous and diverse nascent budding vesicles remains elusive[5,7]. The proposal of a PND-dependent de novo vesicle formation, although not yet independently validated, presents an appealing alternative mechanism alongside the ESCRT-based model for the formation of intraluminal vesicles. Our PND-dependent model of intraluminal vesicle formation not only aligns with our observed results but also offers a potential explanation for how lipids and proteins can be transferred to generate complex MVBs. Further experimental validation is required to substantiate both the ESCRT-based and the hemifusome/PND-based models of intraluminal vesicle formation.

Regardless of their biogenesis—whether by hemifusion, de novo formation driven by PNDs, or another process—these vesicle complexes with two or more compartments separated by HDs expand the diverse repertoire of membrane remodeling in cellular endomembrane systems. The HDs, with their unique bilayer formed by two exoplasmic leaflets, will impact the conformation and distribution of proteins according to their topological sensitivity[50,51]. They could be involved in various processes including protein sorting, lipid transfer[52] and lipid sorting[53], or could explain the existence of lipid raft-like microdomains that retain specific proteins[47]. Independent on how the hemifusomes are formed, if HD expansion occurs at the expense of lipid removal from the outer monolayer[6,38], PNDs may also function as a buffer to mitigate such lipid imbalances. An intriguing possibility is the relationship between hemifusomes and recently reported lysosome-related organelles that contain an expansion compartment mediating zinc transporter delivery[54], although these organelles are micrometer-sized structures and appear much larger than hemifusomes. The

heterotypic vesicles in hemifusomes may also be related to a class of compound vesicular organelles called amphisomes[46].

In situ cryo-ET is arguably the most promising approach for obtaining native structural information on cellular organelles. The overall quality of our images attests to the preservation of structural integrity. For instance, our in situ images of clathrin-coated pits and vesicles (Figs. 1b, 2a, and Supplementary Fig. 2a) reveal additional structural features beyond those visualized using the widely adopted unroofing method[55,56]. In our in situ cryo-ET approach, we aimed to minimize artifacts or stress responses in cells that often result from sample handling and preparation. We achieved this by reducing sample manipulation to essential steps only. Specifically, the process involved a brief 2–3 s transfer of unperturbed cells from the tissue culture medium to the plunge-freezing apparatus, followed by a 4–6 s blotting phase in the humid chamber of the apparatus prior to vitrification. It is thus unlikely that hemifusomes and their associated (PNDs) are sample preparation artefacts. However, if hemifusomes are indeed formed because of this minimal handling, they can be regarded as part of a rapid cellular stress response that must be considered in the evaluation of in situ cryo-ET data. It is more likely that hemifusomes and their HDs were previously overlooked in conventional electron microscopy due to fixation and dehydration steps, which may alter their stability, longevity, or appearance. Our cryo-ET observations have been limited to the thin regions at the cellular periphery. Future research should focus on determining whether hemifusomes and compound hemifusomes are present in other cellular regions and on elucidating the molecular mechanisms underlying their formation, stability, and function, as well as their broader implications for cellular physiology and pathology.

## Methods

### Cell culture and cryo-preparation

Four cell lines were used: COS-7 (ATCC CRL-1651), HeLa (ATCC CCL-2), NIH/3T3 (ATCC CRL-1658), and RAT-1 cells stably expressing claudin2-GFP[57,58]. RAT-1 cells[58] were a gift from James M. Anderson (National Heart, Lung, and Blood Institute, National Institutes of Health). Cells were cultured in Dulbecco's modified Eagle's medium (DMEM) supplemented with GlutaMAX (Thermo Fisher Scientific, 10569010) and 10% heat-inactivated fetal bovine serum (HI-FBS; Thermo Fisher Scientific, 10082147) at 37 °C in a 5% CO₂ humidified incubator. Cells were detached from culture flasks (Falcon, 353109) by rinsing with Dulbecco's phosphate-buffered saline (DPBS; Thermo Fisher Scientific, 14190144) and treating with 0.05% Trypsin-EDTA (Thermo Fisher Scientific, 25300054) at 37 °C for 2–3 min.

For cryo-preparation, gold EM grids with Quantifoil holey carbon film (Quantifoil, R2/1, 300 mesh, gold; Electron Microscopy Sciences, Q3100AR1; or Quantifoil, R3.5/1, 200 mesh, gold; TED PELLA, 660-200-AU-100) were mounted onto PDMS stencil grid holders (Alveole). PDMS stencils attached to coverslips were sterilized under a UV lamp for 2 h. EM grids and stencils were glow-discharged (PELCO easiGlow, TED PELLA) for 30 s at 15 mA and coated with 10 µg/mL fibronectin (Fisher Scientific, 3416351MG) in DPBS for 2 h at room temperature. Cells were seeded on 35 mm tissue culture dishes (Falcon, 353001) containing 2–3 fibronectin-coated EM grids at a density of 100,000 cells per dish for COS-7, NIH/3T3, and RAT-1 cells, and 80,000 cells per dish for HeLa cells. After overnight incubation at 37 °C and 5% CO₂, the medium was replaced with serum-free DMEM before plunge freezing.

Grids were carefully picked up using snap-lock forceps (Leica Microsystems, 16706435) and excess liquid was blotted by touching the edge with a Whatman grade 40 filter paper (Whatman, 1440150). A 1 µL solution of 10 nm colloidal gold coated with BSA in DMEM was added to the grids[59]. Samples were back-blotted for 6 s with Whatman grade 1 filter paper (Leica Microsystems, 16706440) and plunge-frozen into liquid ethane using a Leica EM GP plunger (Leica

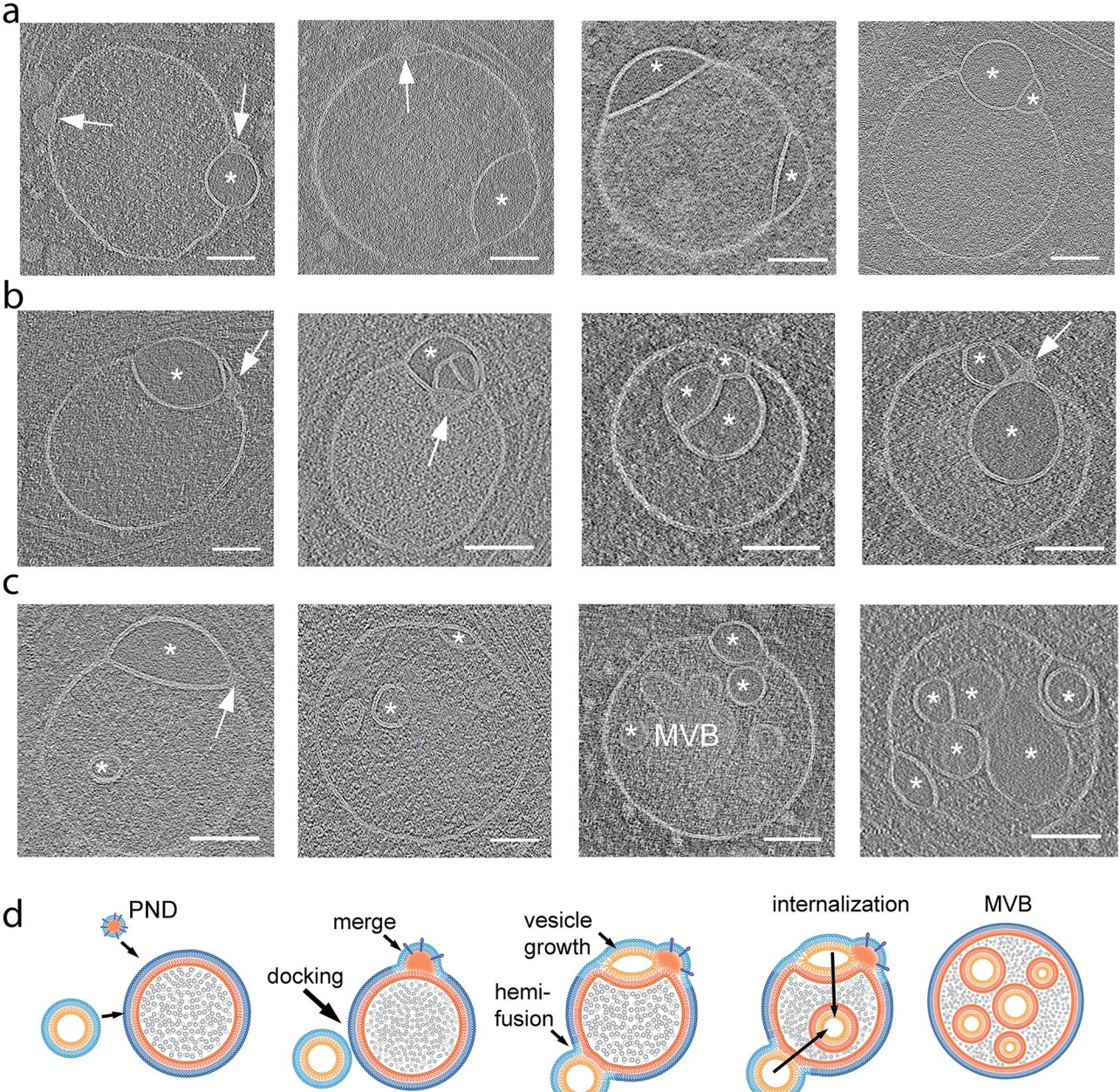

**Fig. 7 | Compound hemifusomes as hubs for the formation of multivesicular bodies. a** Tomographic slices illustrating compound hemifusomes with one or more additional vesicles in a hemifused conformation with either of the two vesicles of the initial hemifusome pair. Asterisks mark vesicles with clear luminal content. Hemifusomes with additional PNDs embedded in their membranes (arrows) suggest these may act as a hub for the formation of compound hemifusomes (*n* = 42 from 308 tomograms). **b** Compound hemifusion with flipped hemifusomes. Multiple hemifusion events coalesce to form very complex compound structures.

Asterisks mark vesicles with clear luminal content. (representative images from at least 5 tomograms). **c** Hemifusomes comprising hemifused vesicles as well as multiple intraluminal vesicles. Asterisks mark vesicles with clear luminal content. Arrow points to PND. (representative images from at least 5 tomograms). **d** Diagram illustrating the proposed path for the formation of compound hemifusomes, followed by the flipping of the vesicle into the luminal side of the larger vesicle and subsequent scission, providing an alternative path to the formation of multivesicular bodies (MVBs). Scale bar: 100 nm.

Microsystems) at 25 °C, 85% humidity, and −176 °C. Samples were stored under liquid nitrogen until data collection.

## Nanogold tracing experiments

Gold nanoparticles used in uptake studies included cell uptake, slightly negatively charged, polymer-functionalized gold nanoparticles (NanoPartz, CO), and transferrin-functionalized gold nanoparticles (Luna Nanotech, ON) as listed below. COS-7 cells were transfected with 1 µg of transferrin receptor mCherry-TFR-20 (Addgene, 55144) using Lipofectamine LTX (Thermo Fisher Scientific, 15338100) overnight. Functionalized nanoparticles were sonicated for 5 min before use and

diluted in fresh DMEM with 10% HI-FBS for uptake experiments. Details of gold nanoparticle concentrations and uptake times are as follows:

- 15 nm Human transferrin covalently functionalized gold nanoparticles with AFDye 647 (GNP-TF-15-H-1-AF6): 1:10 dilution, uptake time: 30 min to 1 h 15 min.
- 15 nm Human transferrin physisorbed gold nanoparticles with AFDye 647 (GNP-TF(PA)−15-H-1-AF6): 1:10 dilution, uptake time: 45 min to 1 h 45 min.
- 5 nm cell uptake polymer-functionalized gold nanoparticles with AFDye 488 (PCU11-5-AF488-NCU-PBS-50-1-CS): 1:50 or 1:200 dilution, uptake time: 1 h to 3 h 15 min.

- 15 nm cell uptake polymer-functionalized gold nanoparticles with AFDye 488 (PCU11-15-AF488-NCU-PBS-50-1-CS): 1:150 or 1:100 dilution, uptake time: 30 min to 5 h.

### Cryo-ET imaging

Vitrified samples were imaged using a 300 keV Titan Krios transmission electron microscope (Thermo Fisher Scientific) equipped with a Bioquantum post-column energy filter (Gatan, Pleasanton, CA) operated in zero-loss mode with a slit width of 20 eV and a defocus range of 2.5–4 μm. Tomo control software (Tomography 5.6.0.2368REL, Thermo Fisher Scientific) was used to record dose-symmetric tilt series[21] from −60° to +60° at either 2.5° or 3° increments. Tilt series images were collected using a K3 Summit direct electron detector (Gatan) at ×27,000 and ×31,000 magnifications under low-dose conditions in counting mode (6 or 10 frames per tilt series image, 0.05 s per frame, 2.16 or 2.69 Å pixel size). The cumulative electron dose per tilt series was limited to under 120 $e^-/Å^2$.

### Data processing

K3 movie frames were dose-weighted and motion-corrected using MotionCor2, then merged using the IMOD software package[60] to generate the final tilt series data. Frames affected by drift or blockage were excluded from reconstruction. Preprocessing and 3D reconstruction were performed using IMOD v4.12.25 or AreTomo[61]. Tilt series images for hemifusion reconstructions were aligned using 10 nm gold nanoparticles as fiducials with IMOD or AreTomo2 for automated marker-free alignment. 3D reconstructions were calculated using weighted back-projection (WBP) algorithm from IMOD or AreTomo2, and simultaneous algebraic reconstruction technique algorithm from AreTomo2 to enhance the contrast. Tomograms were selected based on their effectiveness in accentuating hemifusion characteristics. For visualization of raw tomographic slices, tomograms were denoised using weighted median filter (smooth filter) implemented in IMOD[60] to enhance the contrast for WBP reconstructed tomograms. All tomograms and tomographic slices presented in the figures and supplementary movies are displayed in reverse contrast, unless otherwise specified. We used MemBrain v2 package for the visualization and analysis of cellular membranes in Supplementary Figs. 3, 4, and 7[62]. MemBrain-seg module's automated segmentation and Surface-Dice loss function enhanced the membrane connectivity on the cryo-ET data[62]. Isosurface renderings and curation of the segmentation were performed using UCSF ChimeraX v1.6[63] and Amira v2024.1[64].

### High-performance phase-contrast light microscopy

For live cell imaging cells were plated on glass bottom dish pretreated with 10 μg/mL fibronectin (Fisher Scientific, 3416351MG) in DPBS for 2 h at room temperature. Cells were imaged using a Nikon Ti2 microscope equipped with a Solis® 470 nm High-Power LED illuminator; a 100× CFI60 Plan Apochromat Lambda Phase Contrast DM 100x Oil Immersion Objective Lens, N.A. 1.45; and 4× secondary magnification. The light source was adjusted for oblique illumination for improved resolution and contrast as described previously[65]. Images were acquired with short 30 ms exposures to eliminate blurring of fast-moving organelles and recorded as time-lapse movies.

### Reporting summary

Further information on research design is available in the Nature Portfolio Reporting Summary linked to this article.

## Data availability

The tomograms displayed in Supplementary Movies 1–3 have been deposited in the Electron Microscopy Data Bank (EMDB) under accession numbers EMD-47672, EMD-47671, and EMD-47670, respectively. All other data required to evaluate the conclusions of this study are included within the paper. Cryo-EM data will be submitted to EMPIAR upon publication. Source data are provided with this paper.

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

## Acknowledgements

We thank Dr. Michael Purdy and the UVA Molecular Microscopy facility for expert assistance and the use of the Krios electron microscope. We thank Drs. Bettina Winckler and Ilya Levental from the UVA School of Medicine, and Joshua Zimmerberg from the NIH for critical feedback on the manuscript. This work utilized the computational resources of the NIH HPC Biowulf cluster (http://hpc.nih.gov). We thank Ethan Tyler from

the NIH Medical Arts Branch for graphics. This project was supported by NIDCD-NIH-IRP funds Z01-DC000002 to B.K., S.H., and A.T. and by The Owens Family Foundation and a start-up grant from the Center for Cell and Membrane Physiology, University of Virginia School of Medicine, to S.E.

## Author contributions

The project was conceptualized by B.K. and S.E. Experimental work was performed by S.H. and A.T. Cryo-electron microscopy was done by B.K. Cryo-EM data were processed by A.T. The manuscript was written by B.K. and S.E. All authors assisted in editing the manuscript and contributed to data analysis.

## Funding

## Competing interests

The authors declare no competing interest.
