## [Transparent Peer Review file · Nature Communications]

Hemifusomes and Interacting Proteolipid Nanodroplets Mediate Multi-Vesicular Body Formation

Corresponding Author: Dr Bechara Kachar

Version 0:

Reviewer comments:

Reviewer #1

(Remarks to the Author)

This paper has been previously reviewed for Nature Cell Biology, and now been diverted to Nature communications. It remains an overall impressive collection of tomograms on the morphology of a novel organelle or intermediate, where hemifusion diaphragms are observed between two apposing vesicles. The authors have in their response extensively commented on their interpretation and analysis, which I appreciate. The study has high quality and should be published, even though - as the authors admit - the identity of these structures remains somewhat enigmatic. However, given the overall scientific rigor and implications, I strongly favor now publication.

Reviewer #2

(Remarks to the Author)

As stated in my previous review, I consider this manuscript as outstanding, adding a new dimension to our understanding of intracellular vesicle traffic. While I agree with Ref. 1 that there are many questions that one would like to have answered, I reiterate that the quality of the work is very high, with considerable conceptual importance to the field. The points raised in my original review have all been thoughtfully addressed by the authors, resulting in some editorial changes, and thus I suggest publication in its present form.
